# Neuroprotective Potential of L-Glutamate Transporters in Human Induced Pluripotent Stem Cell-Derived Neural Cells against Excitotoxicity

**DOI:** 10.3390/ijms241612605

**Published:** 2023-08-09

**Authors:** Kanako Takahashi, Yuto Ishibashi, Kaori Chujo, Ikuro Suzuki, Kaoru Sato

**Affiliations:** 1Laboratory of Neuropharmacology, Division of Pharmacology, National Institute of Health Sciences, 3-25-26 Tonomachi, Kawasaki-ku, Kawasaki-city, Kanagawa 210-9501, Japan; ktakahashi@nihs.go.jp (K.T.); chu-773@nihs.go.jp (K.C.); 2Department of Electronics, Graduate School of Engineering, Tohoku Institute of Technology, Miyagi 982-8577, Japan; monogatari555@gmail.com (Y.I.); i-suzuki@tohtech.ac.jp (I.S.)

**Keywords:** excitotoxicity, human induced pluripotent stem cell, neuron, astrocyte, L-glutamate transporter, EAAT1, EAAT2

## Abstract

Human induced pluripotent stem cell (hiPSC)-derived neural cells have started to be used in safety/toxicity tests at the preclinical stage of drug development. As previously reported, hiPSC-derived neurons exhibit greater tolerance to excitotoxicity than those of primary cultures of rodent neurons; however, the underlying mechanisms remain unknown. We here investigated the functions of L-glutamate (L-Glu) transporters, the most important machinery to maintain low extracellular L-Glu concentrations, in hiPSC-derived neural cells. We also clarified the contribution of respective L-Glu transporter subtypes. At 63 days in vitro (DIV), we detected neuronal circuit functions in hiPSC-derived neural cells by a microelectrode array system (MEA). At 63 DIV, exposure to 100 μM L-Glu for 24 h did not affect the viability of neural cells. 100 µM L-Glu in the medium decreased to almost 0 μM in 60 min. Pharmacological inhibition of excitatory amino acid transporter 1 (EAAT1) and EAAT2 suppressed almost 100% of L-Glu decrease. In the presence of this inhibitor, 100 μM L-Glu dramatically decreased cell viability. These results suggest that in hiPSC-derived neural cells, EAAT1 and EAAT2 are the predominant L-Glu transporters, and their uptake potentials are the reasons for the tolerance of hiPSC-derived neurons to excitotoxicity.

## 1. Introduction

Currently, a low number of new drugs are within the final steps in the drug development process, which requires approximately 10 years of research and more than USD 800 million [1]. The low success rate is partly due to safety/toxicity problems. Human induced pluripotent stem cell (hiPSC) technology is expected to provide a breakthrough to solve this problem. In 2006, Yamanaka et al. developed iPSC technology, in which adult somatic cells were reprogrammed to pluripotent stem-like cells through the introduction of four genes (OCT4, SOX2, KLF4, MYC) [2]. These iPSC-derived tissue-specific cells are expected to bridge animal models and human tissues with multifactorial and multigenic features. Because one-third of safety issues in clinical trials were attributed to central nervous system (CNS)-related problems [3,4,5], many industrial users (pharmaceuticals, etc.) need the CNS safety/toxicity assessment systems with high human predictability. To meet such needs, hiPSC-derived neural cells have started to be used in safety/toxicity assessments [6].

In the CNS, there is a neuron-specific cell death mechanism called excitotoxicity, in which excessive concentrations of the extracellular excitatory neurotransmitter L-glutamate (L-Glu) activate N-methyl-D-aspartate receptors (NMDARs) and cause acute Ca^2+^ influx, leading to neuronal death [7]. The machinery necessary to reproduce excitotoxicity should therefore be installed in the CNS safety/toxicity assessment systems. To date, most in vitro preclinical studies have employed a primary culture of rodent neural cells or their equivalent cell line systems, in which excitotoxiciy is reproduced by exposure to L-Glu at micromolar concentrations [8]. On the other hand, even when exposed to much higher concentrations of L-Glu, the excitotoxicity levels remain small in hiPSC-derived neural cell cultures regardless of laboratory-made or commercially available [9,10,11]. In this study, we focused on L-Glu transporter functions in hiPSC-derived neural cells. Under physiological conditions, L-Glu transporters (excitatory amino acid transporters: EAATs in humans) mainly expressed in astrocytes rapidly remove extracellular L-Glu in synaptic transmission and protect neurons from excitotoxicity [12,13]. So far, little information has been available concerning the role of L-Glu transporter functions of hiPSC-derived neural cell cultures. In a study using astrocytes prepared from acutely resected surgical tissue, human astrocytes were found to be larger, structurally more complex, and more diverse than those of rodents [14], suggesting the difference of neuron–astrocyte interactions in hiPSC-derived neural cell cultures and in rodent primary cultures. In the present study, taken the practical drug development processes into account, we employed commercially available hiPSC-derived neural cells. We examined whether astrocytes were differentiated and whether EAATs helped remove extracellular L-Glu in this culture system. Here, we showed that functional astrocytes were also differentiated in commercially available hiPSC-derived neural cell culture. We detected the expression of EAAT1 in astrocytes and of EAAT2 in both neurons and astrocytes. We also clarified that these transporters were potent enough to remove exogenously-applied L-Glu immediately; that is the reason for the tolerance of hiPSC-derived neurons to excitotoxicity.

## 2. Results

### 2.1. Functional Maturation of hiPSC-Derived Neural Networks

We first investigated the functional neuronal maturation of hiPSC-derived neural cells over time in culture using a microelectrode array (MEA) system. At 14 days in vitro (DIV), any network bursts were not observed. At 28 DIV, network bursts with very low firing densities were observed for only some electrodes in the raster plot. On the other hand, at 56 DIV, a high frequency of network bursts with high synchronous and high firing density was observed at all electrodes (Figure 1A(a1)). These results indicate that neural networks were formed at 28 DIV at the latest. The difference in firing density between 28 DIV and 56 DIV was more evident when observed with the array wide spike detection rate (AWSDR, Figure 1A(a2)). Four parameters, i.e., number of wells with network activity (Figure 1A(a3)), total spikes per min (Figure 1A(a4)), number of network bursts (NBs) per min (Figure 1A(a5)), and spikes in an NB (Figure 1A(a6)), were compared to examine functional maturation over time between 14, 21, 28, 42, and 56 DIV. NBs were detected after 28 DIV, and the number of wells with network activity increased with culture days. Total spikes per min, number of NBs per min, and spikes in an NB increased with culture days. These results further support that in hiPSC-derived neural cell culture, neural networks began to form at approximately 28 DIV and reached functional maturation by 56 DIV.

We also confirmed synaptic maturation at 63 DIV by immunocytochemical experiments (Figure 1B). The presynaptic protein synapsin 1 (Syn1) puncta merged with vesicular glutamate transporter 2 (Vglut2) puncta (white arrowheads in Figure 1B(b1-2)). Pearson’s correlation coefficient (PCC), the parameter indicating the co-localization between Vglut2 and Syn1 were increased with culture days (Figure 1B(b1-3)), showing the maturation of the glutamatergic pre-synapse structure. Syn1 puncta were closely adjoined postsynaptic density protein 95 (PSD95) puncta (yellow arrowheads in Figure 1B(b2-2)). PCC between PSD95 and Syn1 at DIV 63 were low (0.284 ± 0.029, *n* = 3), indicating that PSD95 is contiguous to Syn1 to be more precise. These changes in localization pattern are consistent with previous reports [15] and indicate the synaptic maturation. These results indicate that glutamatergic synaptic structures were mature at 63 DIV. Taking the MEA data and the immunocytochemical data together, we judged that matured neural circuits had been formed by 63 DIV in hiPSC-derived neural cell culture.

### 2.2. Astrocytes Are Differentiated in hiPSC-Derived Neural Cell Cultures

We examined the expression of markers for stem cells, neurons, and astrocytes at the mRNA and protein levels at 0 (just before seeding), 14, and 63 DIV. The gene expression of *NESTIN*, a neuronal stem cell marker [16], was decreased with the culture period (Figure 2A(a1)). The gene expression of *HuC/D*, a neuronal marker [17], was transiently increased at 14 DIV and then slightly decreased at 63 DIV (Figure 2A(a2)). On the other hand, the gene expression levels of *GFAP* and *S100β*, astrocyte markers [18,19], were increased with the culture period (Figure 2A(a3,a4)). When we performed Western blotting analysis (Figure 2B) and immunocytochemistry (Figure 2C), similar tendencies were observed. As shown in Western blotting data, the expression level of HuC/D was slightly decreased with the culture period (Figure 2B(b1,b2)). Immunocytochemical data showed HuC/D+ neurons at 14 and 63 DIV (Figure 2C). On the other hand, the expression of GFAP and S100β was clearly increased with the culture period (Figure 2B(b1,b3,b4)) in Western blotting. In the immunocytochemical data, an increase in the numbers of GFAP+ and S100β+ cells were clearly observed (Figure 2C). These results indicate that in hiPSC-neural cell culture, astrocytes are also differentiated, while stemness is decreased with the culture period.

### 2.3. The Expression of EAAT1 and EAAT2 in hiPSC-Derived Neural Cells

In preliminary experiments, we performed qRT-PCR of EAAT1-5 using 63 DIV hiPSC-derived neural cell culture and confirmed no expressions of EAAT4 and 5. Although mRNA expression of EAAT3 was detected in qRT-PCR, the protein expression level of EAAT3 was very low compared to EAAT1 and 2. On the other hand, qRT-PCR analysis showed that the gene expressions of EAAT1 (Figure 3A(a1)) and EAAT2 (Figure 3A(a2)) were clearly increased during the culture period and that the expression of EAAT2 was more acutely increased than that of EAAT1. We also confirmed that the protein expression levels of EAAT1 and EAAT2 were culture period-dependent by Western blotting (Figure 3B). We then investigated the cell types that expressed EAAT1 (Figure 3C(c1)) or EAAT2 (Figure 3C(c2)) at 63 DIV by immunocytochemistry. We determined the cell types based on the expression patterns of marker proteins, i.e., GFAP+Nestin+ for radial glial cells, GFAP+S100β+ for astrocytes, and HuC/D+MAP2+ for neurons. Representative images of the immunocytochemistry are shown in Figure 3C. EAAT1 was mainly expressed in radial glia and astrocytes (Figure 3C(c1)), while EAAT2 was expressed in radial glia, astrocytes, and neurons (Figure 3C(c2)). 

### 2.4. The Roles of EAATs in the Sensitivity of hiPSC-Derived Neurons to Excitotoxicity

When the extracellular L-Glu concentration is elevated beyond control, Ca^2+^ overload is induced in neurons through NMDAR and neuronal death, which is called ‘excitotoxicity’ [20,21,22,23]. Excitotoxicity has been reported to be the critical step in various kinds of CNS disorders [24,25]. In vitro neural cultures in preclinical studies are therefore desirable for installing the machinery necessary for reproducing excitotoxicity. Although hiPSC-derived neural cell cultures have been developed for translational studies between preclinical and clinical studies, the information concerning the sensitivity of these cells to excitotoxicity is insufficient. We first examined the effects of 24 h application of 100 µM L-Glu, the typical experimental condition that causes almost 100% neuronal death in primary culture of rodent neurons [26,27] at 14 DIV and 63 DIV. The cell viability was quantified by the extent of MTT reduction [28]. As shown in Figure 4A(a1,a2), a 24 h application of 100 μM L-Glu generated no effects on the cell viabilities at both 14 and 63 DIV. L-Glu transporters remove the excitatory neurotransmitter L-Glu from the synaptic cleft just after release and maintain extracellular L-Glu concentration homeostasis [12,13]. We therefore examined the contribution of EAATs to the tolerance to high concentrations of L-Glu. TFB-TBOA (TFB) blocks EAAT1-3 and mainly blocks EAAT1 and 2 at 30 nM (Appendix A) [29]. When 30 nM TFB-TBOA was coapplied with 100 μM L-Glu for 24 h at 14 DIV, it caused no additive effects on L-Glu alone (Figure 4B(b1)). In contrast, at 63 DIV, co-application of TFB and L-Glu significantly decreased MTT reduction compared with that of L-Glu alone (Figure 4B(b2)). AP5 (100 µM), an NMDAR antagonist, almost completely abrogated the effects of TFB (Figure 4B(b2)). These results indicate that L-Glu transporters are active in hiPSC-derived neural cell culture at 63 DIV; thus, we investigated the uptake potential of L-Glu transporters in this culture. We measured the extracellular concentrations of L-Glu at 0 min, 10 min, 20 min, 30 min, and 60 min after application at 100 µM (Figure 4C(c1)). The L-Glu concentration started to decrease immediately after the application, reaching 14.8 ± 1.0 µM (*n* = 3) 60 min after. We also investigated the contribution of each EAAT subtype pharmacologically using 100 µM UCPH-101 (UCPH, a selective EAAT1 inhibitor) [30], 300 µM dihydrokainic acid (DHK, a competitive EAAT2 inhibitor) [31], 30 nM TFB, and 10 µM WAY213613 (WAY, a potent competitive EAAT2 inhibitor with inhibition of EAAT1 and EAAT3 at 10 µM) [32]. The IC50 values for EAAT1-3 of these compounds are shown in Appendix A. Figure 4C(c2) compares the inhibitory potentials of these inhibitors at 30 min of L-Glu uptake, and a 50% decrease in extracellular L-Glu was observed (Figure 4C(c1)). The inhibition of UCPH, DHK, TFB, and WAY was 39.4 ± 18.6% (*n* = 5), 48.9 ± 11.0% (*n* = 5), 89.5 ± 9.7% (*n* = 5), and 80.5 ± 17.5% (*n* = 7), respectively. The effect of DHK was slightly stronger than that of UCPH, while the effect of TFB was much stronger than those two inhibitors. Because the effect of WAY is slightly weaker than that of TFB, the contribution of EAAT3 was considered to be small. These results strongly suggest that extracellular L-Glu was removed mainly by EAAT1 and EAAT2. We further confirmed the contributions of the L-Glu transporter subtypes to L-Glu uptake in this hiPSC-derived neural cell culture by analyzing the correlation between the inhibitory potentials and the decrease in cell viabilities (MTT reductions) (Figure 4D). The blue (UCPH), green (DHK), pink (WAY), and red (TFB) dots were distributed from left to right. The inverse correlation between the inhibitory potentials and the decrease in viabilities was significant, as shown by Pearson’s correlation coefficient (PCC = −0.7110), supporting that EAAT1 and EAAT2 play main roles in preventing excitotoxicity. In addition, the contribution of EAAT2 tended to be stronger than that of EAAT1.

## 3. Discussion

### 3.1. Patterns of the Expression of Developmental Markers and L-Glu Transporters

In hiPSC-derived neural cell culture, the expression of HuC/D was increased at first, followed by an increase in the expression of GFAP and S100β, while the expression of nestin was gradually decreased. The gliogenesis of protoplasmic astrocytes is a late event in the human fetal period that occurs after neurogenesis [33], in which process radial glial cells play main roles [34,35]. These results suggest that the cellular differentiation pattern in these hiPSC-derived neural cells follows CNS developmental steps. In vivo observations indicated that EAAT1 protein is expressed in radial glia in the periventricular zone [36] and in GFAP-positive astrocytes in the cerebral cortex [37], while EAAT2 protein is expressed in both neurons and astrocytes in the cerebral cortex [37]. The expression patterns of EAAT1 and EAAT2 were also consistent with those of the in vivo forebrain. Taken together, the results indicate that the hiPSC-derived neural cells used in this study are differentiated following the physiological developmental process.

### 3.2. EAAT Subtypes Contribute to Tolerance to Excitotoxicity in hiPSC-Derived Neurons

EAATs have five subtypes: EAAT1, 2, 3, 4, and 5. Our data show that EAAT1 and EAAT2 are the main L-Glu transporters that remove extracellular L-Glu in hiPSC-derived neural cell culture. It has been reported that when astrocytes are differentiated from iPSCs, over 2–3 months are needed to express EAAT1 [38,39,40]. Our data indicate that the astrocytes differentiated in the iPSC-derived neural cell culture mainly express EAAT1 and EAAT2 at 63 DIV, and these L-Glu transporters protect neurons from excitotoxicity caused by 100 µM L-Glu, the experimental condition that has been routinely used to cause almost 100% cell death in primary culture of rodent neurons [26,27].

In this study, we detected EAAT1 expression mainly in astrocytes, while EAAT2 was expressed in both astrocytes and neurons. Rodent studies have shown that the protein expression levels of GLAST (EAAT1 in humans) and GLT1 (EAAT2 in humans) are dramatically increased during synaptogenesis beginning from E18 [41]. Although GLAST and GLT1 are astrocytic L-Glu transporters, GLT1 was also reported to be expressed in neurons [42]. A study using synaptosomes showed that neuronal GLT1 significantly contributes to L-Glu uptake [43]. Therefore, the expression pattern of EAAT1 and EAAT2 in hiPSC-derived neural cells is consistent with that in vivo. Furthermore, although GLT1 (EAAT2) accounts for ~90% of L-Glu uptake in the forebrain [44], cultured pure astrocytes express only GLAST (EAAT1) [43]. On the other hand, because the expression of both GLAST (EAAT1) and GLT1 (EAAT2) was detected in mixed cultures of astrocytes and neurons [43], the expression of GLT1 (EAAT2) is suggested to be regulated by soluble factors from neurons [45,46,47,48]. These mechanisms may be active in this hiPSC-derived neural cell culture, as well.

### 3.3. The Significance of Our Data for the CNS Safety/Toxicity Assessment System at Preclinical Stage

Before human stem cell-derived neurons appeared, in vitro toxicity and safety tests were performed using rodent primary cultures [49,50,51,52]. In these primary neuron cultures, exposure to 100 µM L-Glu caused severe excitotoxicity, thereby leading to almost 100% neuronal death in both neuronal culture and neuron–glia coculture. To date, although excitotoxicity is reproduced in hiPSC-derived neurons in many reports, the concentrations of L-Glu used are very high (>1 mM), and the resulting cell damages are very small [9,10,11]. According to a study using astrocytes prepared from acutely resected surgical tissue, human astrocytes were found to be larger, structurally more complex, and more diverse than those of rodents [14], suggesting that neuron–astrocyte interaction in hiPSC-derived neural cell culture is one of the causes of the neuronal tolerance to excitotoxicity; however, little information is available. In our data, when L-Glu transporters (mainly EAAT1 and 2) were inhibited, 100 µM L-Glu induced a significant decrease in the viability of hiPSC-derived neural cell culture. We also confirmed that the decrease in cell viability was mediated by NMDAR. We clarified that hiPSC-derived neural cell cultures are surely installed with the machineries to reproduce excitotoxicity, and the L-Glu transporters in both astrocytes and neurons strongly protect neurons from excitotoxicity. Furthermore, our data also suggest that this hiPSC-derived neural cell culture is useful for screening drugs targeting L-Glu transporters, which had been difficult because of the low expression level of EAAT2 in rodent primary cultures [53]. We should avoid discussing species differences here based on these in vitro data because the neuron–glia composition and developmental stage of each cell type are completely different between rodent primary cultures and hiPSC-derived neural cell cultures. From our data, we can conclude that the hiPSC-derived neural cell cultures provide useful in vitro CNS safety/toxicity assessment systems and L-Glu transporter screening systems that have the same molecular machineries as human CNS cells.

## 4. Materials and Methods

### 4.1. Materials

All chemical compounds were purchased from FUJIFILM Wako Pure Chemical (Oosaka, Japan) unless otherwise stated. Protease inhibitor cocktail set1 (539131) was purchased from Calbiochem (Darmstadt, Land Hessen, Germany). Can Get Signal^TM^ was purchased from Toyobo (Osaka, Japan). Stripping buffer was purchased from Thermo Fisher Scientific (Waltham, MA, USA). Bromophenol blue sodium salt (BPB), bovine serum albumin (BSA), dimethyl sulfoxide (DMSO), β-nicotinamide adenine dinucleotide (β-NAD), dihydrokainic acid (DHK, an EAAT2-specific inhibitor), 3-(4,5-dimethyl-2-thiazolyl)-2,5-diphenyl-2H-tetra-zolium bromide (MTT), 1-methoxy-5-methyl-phenazinium methyl sulfate (MPMS), and sodium dodecyl sulfate (SDS) were purchased from Sigma-Aldrich (Darmstadt, Germany). (3s)-3-[[3-[[4-(trifluoromethyl)benzoyl]amino]phenyl]methoxy]-l-aspartic acid (TFB-TBOA, TFB, a nonspecific EAAT inhibitor), D-(-)-2-amino-5-phosphonopentanoic acid (AP5, NMDAR antagonist), and N-[4-(2-bromo-4,5-difluorophenoxy)phenyl]-L-asparagine (WAY213613, a potent EAAT2 inhibitor) were purchased from TOCRIS (Minneapolis, USA). 2-Amino-5,6,7,8-tetrahydro-4-(4-methoxyphenyl)-7-(naphthalen-1-yl)-5-oxo-4H-chromene-3-carbonitrile (UCPH-101, UCPH, an EAAT1-specific inhibitor) was purchased from Abcam (Cambridge, UK). Bovine liver glutamate dehydrogenase (GLUD1) was purchased from Roche (Mannheim, Germany). Tris (hydroxymethyl) aminomethane (Tris–HCl) was purchased from Bio-Rad (Hercules, CA, USA). Ethylenediaminetetraacetate (EDTA) and ethylene glycol tetraacetate (EGTA) were purchased from Dojindo (Kumamoto, Japan). Goat serum was purchased from Vector Laboratories (Newark, CA, USA). Donkey serum was purchased from Rockland Immunochemicals (Boyertown, PA, USA).

### 4.2. Culture of hiPSC-Derived Neurons

Commercially available hiPSC-neurons were used in this study (XCL-1 neurons, XCell Science, Novato, CA, USA). The manufacturer’s instruction explains that these cells were made according to the protocol for cortical neurons. We confirmed the expression of FOXG1, the marker of the frontal cortex in these cells. The hiPSC-neurons were cultured according to the manufacturer’s instructions with modifications. Cells were plated in 48-well plastic plates, 8-well glass chambers (155409JP, Nunc, Worcester, MA, USA), 16 channels per well across 24-well MEA plates (eco24, MED-Q2430M, Alpha Med Scientific, Oosaka, Japan), or 96-well plastic plates at a density of 3.0 × 10^5^ cells/cm^2^ in neural maturation basal medium (NM-001-BM100, XCell Science) with neuron maturation supplement A (NM-001-SA100, XCell Science). Plates were precoated with polyethyleneimine (PEI) (for MEA, 0.005% for 10 min at 37 °C; for others, 0.025% for 1 h at room temperature; P3413, Sigma-Aldrich) and then coated with Matrix-511 (2.5 µg/mL for 3 h at 37 °C, 892011, Nippi, Tokyo, Japan). For WB, ICC, and MEA cultures, a glass ring with a diameter of 3.4 mm (Ring-05, Iwaki, Shizuoka, Japan) was placed in the center of the well, and cell suspensions were seeded in the ring. Half of the medium was replaced every 2 days. After 8 days of culture, the medium was replaced with BrainPhys^TM^ Neuronal medium with SM1 neuronal supplement (STEMCELL Technologies, Vancouver, Canada). Half of the medium was replaced every 4 days. All experiments using hiPSC-neurons were approved by the Research Ethics Committee of National Institute of Health Sciences (NIHs) in accordance with the Declaration of Helsinki.

### 4.3. Quantitative Real-Time Reverse Transcription Polymerase Chain Reaction (qRT-PCR)

Total RNA was isolated from cells using TRIzol reagent (Sigma-Aldrich). The amount of total RNA was quantified by measuring the OD260 using a Nanodrop spectrophotometer (Nanodrop, Thermo Fisher Scientific). Real-time PCR was performed using the QuantiTect SYBR Green RT-PCR kit (Qiagen, Hilden, Germany) and an ABI PRISM 7900HT sequence detection system (Thermo Fisher Scientific) according to the manufacturer’s protocol. The reactions (20 μL) contained 5 ng of total RNA and 0.5 μM forward and reverse primers in the master mix solution. The data were analyzed with 7900 System SDS Software 2.2.2 (Thermo Fisher Scientific) by relative quantification using the comparative C_T_ method. Relative changes in transcript levels were normalized to the mRNA levels of β-Actin.

The primer sequences were as follows: 5′-CCAAGACTGCCCTGGAAAC-3′, 5′-CCTCCCTCTCCAAGGAAACA-3′ (NESTIN), 5′-CCTCAAATTACAGACGAAGACCA-3′, 5′-GCTGACGTACAGGTTAGCATC-3′ (HuC/D), 5′-GCCATTGCCTCATACTGCGT-3′ (GFAP), 5′-TGGCCCTCATCGACGTTTTC-3′, 5′-ATGTTCAAAGAACTCGTGGCA-3′ (S100β), 5′-CATGTACGTTGCTATCCAGGC-3′ 5′-CTCCTTAATGTCACGCACGAT-3′ (β-Actin), 5′-ATGAGGATGTTACAGATGCTGG-3′ 5′-CAGGATGGATGATGATGACAAT-3′ (EAAT1), 5′-CTGTTGTCTCTCTGTTGAACG-3′, 5′-CAACCACTTCTAAGTCCTTGATTG-3′ (EAAT2).

### 4.4. Western Blotting

The cells were lysed with NP-40 lysis buffer [150 mM NaCl, 10 mM EDTA, 5 mM EGTA, 0.5% NP-40, and 0.5% sodium deoxycholate in 10 mM Tris–HCl buffer (pH 6.8)]. The protein concentration was measured using a BCA protein assay (Pierce™ BCA Protein Assay Kit, 23225, Thermo Fisher Scientific). The proteins (10 μg/lane) were resolved with SDS—PAGE and transferred onto a PVDF membrane (Bio-Rad, Hercules, CA, USA). The membrane was blocked with 3% nonfat dry milk or Block Ace blocking solution (DS Pharma Biomedical, Osaka, Japan) for 1 h at room temperature. The membrane was incubated with mouse anti-HuC/D monoclonal antibody (1:100, A21271, Thermo Fisher Scientific), mouse anti-GFAP monoclonal antibody (1:5000, MAB3402, Millipore, Hessen, Germany), rabbit anti-S100β polyclonal antibody (1:5000, A5971, Sigma-Aldrich), rabbit anti-EAAT1 polyclonal antibody (1:1000, 5684, Cell Signaling, Beverly, MA, USA), guineapig anti-EAAT2 polyclonal antibody (1:5000, AB1783, Chemicom), or mouse anti β-actin monoclonal antibody (1:5000, ab8226, Abcam) overnight at 4 °C, followed by incubation with horseradish peroxidase-conjugated anti-mouse, anti-rabbit (1:20,000, Amersham Biosciences, Buckinghamshire, UK), or guineapig (1:50,000, Invitrogen, Waltham, MA, USA) antibodies. The signals were scanned with an LAS3000 (Fujifilm, Tokyo, Japan) using an ECL Western blot detection system (SuperSignal^TM^ West Femto maximum Sensitivity Substrate, 34095, Thermo Fisher Scientific). Relative densities of bands were quantified using Multi Gauge software (FUJIFILM). Relative changes in expression were determined by normalization to β-actin. Representative data are shown in Figures, and the expression levels are normalized to those of 14 DIV. The same results were obtained in four independent experiments.

### 4.5. Immunocytochemistry Scale

The cells were fixed with 4% PFA for 1 h at room temperature. After a PBS wash was performed, the cells were immunostained using the AbScale clearing/labeling protocol [54]. The fixed cells were permeabilized and cleared with sequential incubation in multiple solutions: ScaleS0, ScaleA2, ScaleB4(0), and ScaleA2. Then, after deScaling by washing with PBS, the cells were incubated with primary antibodies for 3 days at 4 °C in an AbScale solution. Mouse anti-HuC/D monoclonal antibody (1:100, A21271, Thermo Fisher Scientific), chicken anti-GFAP polyclonal antibody (1:400, ab4674, Abcam), rabbit anti-S100β polyclonal antibody (1:500, ab52642, Abcam), goat anti-Vglut2 polyclonal antibody (1:500, Go-Af310-1, Frontier Institute, Hokkaido, Japan), rabbit anti-Syn1 polyclonal antibody (1:1000, AB1543, Chemicon), chicken anti-MAP2 polyclonal antibody (1:5000, ab5392, Abcam), mouse anti-PSD95 monoclonal antibody (1:500, 7E3-1B8, Thermo Fisher Scientific), mouse anti-EAAT1 monoclonal antibody (1:200, ab49643, Abcam), rabbit anti-Nestin polyclonal antibody (1:200, ABD69, Millipore), or guineapig anti-EAAT2 polyclonal antibody (1:1000, AB1783, Chemicon) were used. The cells were incubated with secondary antibodies conjugated to fluorochromes (1:500, Invitrogen) and Hoechst (1:200, Dojindo, Kumamoto, Japan) for 2 days at 4 °C. Before refixation with 4% PFA, cells were rinsed in an AbScale rinse solution. Finally, cells were optically cleared by incubation in ScaleS4. The composition of the solution was as follows: ScaleS0 (a PBS(−) solution containing 20% D-(−)-sorbitol, 5% glycerol, 1 mM methyl-β-cyclodextrin, 1 mM γ-cyclodextrin, 1 mM N-acetyl-L-hydroxyproline, 3% DMSO, pH 7.2), ScaleA2 (10% glycerol, 4 M urea, 0.1% Triton X-100, pH 7.7), ScaleB4(0) (8 M urea, pH 8.4), AbScale (a PBS(−) solution containing 0.33 M urea, 0.1–0.5% Triton X-100, pH 7.2), AbScale rinse solution (a 0.1× PBS(−) solution containing 2.5% BSA, 0.05% (*w*/*v*) Tween-20), and ScaleS4 (40% D-(−)-sorbitol, 10% glycerol, 4 M urea, 15–25% DMSO, pH 7.9). The imaging data were collected and analyzed by a Nikon A1R-A1 confocal microscope system (Nikon, Tokyo, Japan).

### 4.6. MEA Recording and Data Analysis (Extracellular Recording, Burst Analysis)

Spontaneous extracellular field potentials were acquired at 37 °C under a 5% CO_2_ atmosphere using a 24-well MEA system (Presto, Alpha Med Scientific, Oosaka, Japan) at a sampling rate of 20 kHz/channels. Signals were high-pass filtered at 0.1 Hz and stored on a personal computer. The spikes in the acquired data were detected using the 100 Hz high-pass filter. Spontaneous firing was recorded at 14 DIV, 21 DIV, 28 DIV, 42 DIV, and 56 DIV. Electrophysiological activities were analyzed using Presto software (Alpha Med Scientific, Oosaka, Japan). A spike was counted when the extracellularly recorded signal exceeded a threshold of ±5.3 σ, where σ was the standard deviation of the baseline noise during quiescent periods. Network bursts (NBs) were detected using the 4-step method, which was described previously [55]. First, spikes separated by interspike intervals of 10 ms were attributed to the same NB. Second, datasets with a maximum number of spikes in the NB below three spikes/NB were eliminated from the analysis. Third, NBs separated by inter-NB intervals shorter than 300 ms were combined. Finally, an NB was defined when it contained more than 50 spikes/NB.

### 4.7. MTT Reduction Assays

Cell viability was determined by MTT reduction activity. MTT reduction activity was measured according to a previously described method [28,53]. Briefly, hiPSC-neurons cultured in the presence or absence of EAAT inhibitor were exposed to L-Glu (100 μM) for 24 h. MTT was added to each well at 15 μg and incubated for 15 min at 37 °C. The medium in each well was carefully removed, and 50 µL of DMSO was added to dissolve the reaction product (MTT formazan). The amount of MTT formazan was determined by measuring the absorbance at 570 nm (test wavelength) and 655 nm (reference wavelength) with an iMark^TM^ microplate reader (Bio-Rad).

### 4.8. Measurement of the Extracellular L-Glu Concentration (L-Glu Uptake Assay)

The L-Glu concentration in the medium was measured according to a previously described method [53]. The medium in the 96-well plates was replaced with fresh medium containing 100 μM L-Glu. Twenty-five microliters of the medium per well was collected. The L-Glu concentration was measured by mixing the medium with 25 μL of substrate mixture [20 U/mL GLUD1, 2.5 mg/mL β-NAD, 0.25 mg/mL MTT, 100 μM MPMS, and 0.1% Triton X-100 in 0.2 M Tris–HCl buffer (pH 8.2)] and incubating at room temperature for 7 min. The reaction was stopped by adding 50 μL of stop solution [50% dimethylformamide and 20% SDS in purified water (DIRECT-Q, Millipore), pH 4.7]. The amount of MTT formazan was determined by measuring the absorbance at 570 nm (test wavelength) and 655 nm (reference wavelength) with an iMark^TM^ microplate reader (Bio-Rad). The extracellular L-Glu concentration was estimated from a standard curve constructed for each assay using cell-free medium containing known concentrations of L-Glu.

### 4.9. Drug Treatment

Stock solutions of 100 mM L-Glu, 25 mM DHK, and 100 mM AP5 in purified water and 50 mM TFB, 25 mM UCPH, and 100 mM WAY in DMSO were dissolved into the medium at the time of application. EAAT inhibitor and AP5 were incubated with the cells for 1 h before the application of L-Glu (100 μM).

DMSO was used as a vehicle control. For the MTT reduction assay and L-Glu uptake activity assay, 0.4% DMSO, the maximum concentration used for drug dilution, was added to all test solutions.

### 4.10. Statistical Analysis

All data are expressed as the mean ± standard deviation. Statistical analyses were performed using Tukey’s test following one-way measures analysis of variance (ANOVA), the Holm–Bonferroni Method following two-way ANOVA, or Student’s *t*-test in GraphPad Prism (GraphPad Software, San Diego, CA, USA), as shown in the figure legends. In all the comparisons, the differences were considered statistically significant when *p* < 0.05. All the experiments were repeated in over triplicate, and the same results were obtained in all sessions.

The Pearson’s correlation coefficients (PCCs) were used to quantify the overlapping fluorescence signals (Figure 1) and the correlation between the strength of L-Glu uptake inhibition and the cell viability caused by EAAT inhibitors (Figure 4). The correlation coefficient represents a number between −1 and 1 that represents the strength and direction of the relationship between two variables. The PCCs for describing reciprocal colocalization between Vglut2 or PSD95 and Syn1 were carried out using NIS-Elements (Nikon, Tokyo, Japan). Max intensity images obtained using a 40× objective were analyzed. The PCCs for the correlation for the effects caused by EAAT inhibitor were determined by linear regression analysis using GraphPad Prism.

## Figures and Tables

**Figure 1 ijms-24-12605-f001:**
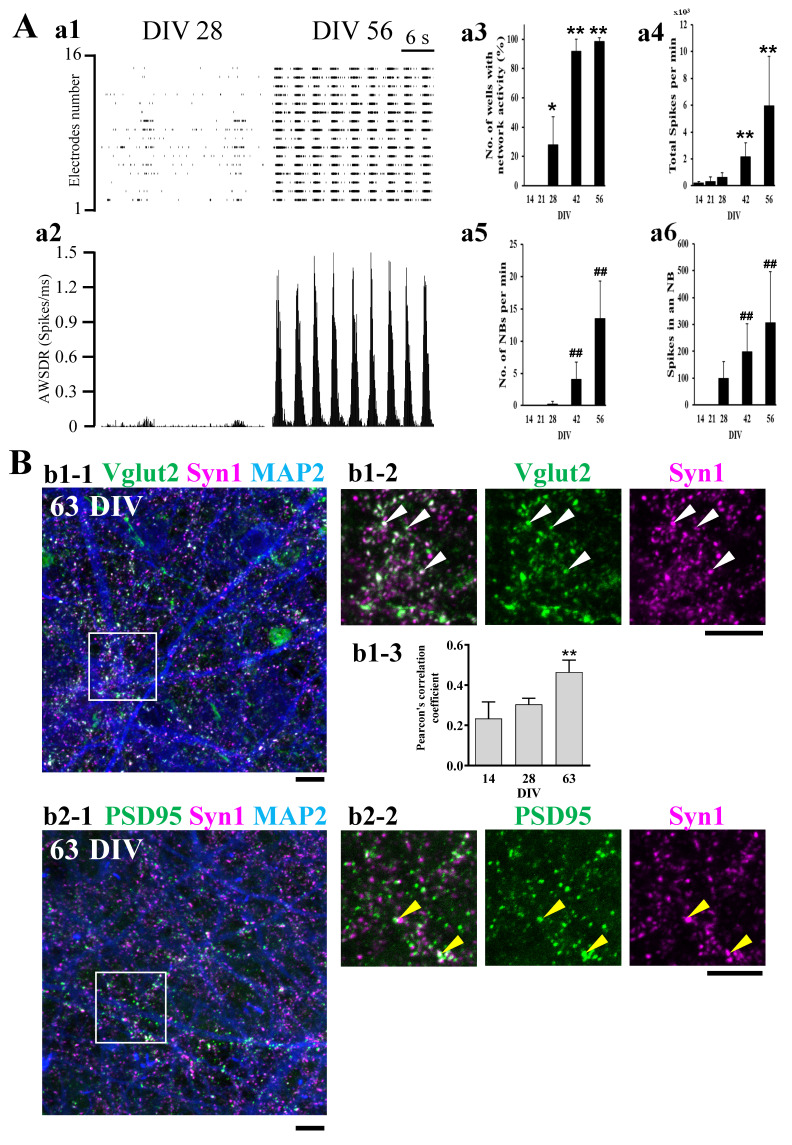
Functional maturation of hiPSC-derived neural networks. (**A**) Spontaneous firing of hiPSC-derived neural networks and comparison of 4 parameters at 14, 21, 28, 42, and 56 DIV (*n* = 72). The measurements were performed for 15 min. Raster plots for all 16 electrodes at 28 and 56 DIV (**a1**). Array wide spike detection rate (AWSDR, number of spikes/ms) at 28 and 56 DIV (**a2**). Network bursts (NBs) were detected after 28 DIV, and the number of wells with network activity increased with culture days (**a3**). Total spikes per min increased with culture days (**a4**). The number of network bursts (NBs) per min increased with culture days (**a5**). Spikes in an NB increased with culture days (**a6**). * *p* < 0.05, ** *p* < 0.01 vs. DIV 14 group and ^##^
*p* < 0.01 vs. DIV 28 group by the Holm–Bonferroni Method following two-way ANOVA. Data are expressed as the means ± standard deviations. (**B**) At 63 DIV, hiPSC-neurons were immunostained with anti-Vglut2 (green: vesicular glutamate transporter 2), anti-Syn1 (magenta: pre-synapse), and anti-MAP2 (blue: dendrite) antibodies (**b1-1**). Magnified view in the white square in b1 (**b1-2**). White arrowheads indicate co-localization of Vglut2 and Syn1 on MAP2-positive fibers. Pearson’s Correlation Coefficient between Vglut2 (green) and Syn1 (magenta) increased with culture days (**b1-3**). At 63 DIV, hiPSC-neurons were immunostained with anti-PSD95 (green: post-synapse), anti-Syn1 (magenta: pre-synapse), and anti-MAP2 (blue: dendrite) antibodies (**b2-1**). Magnified view in the white square in b2 (**b2-2**). Yellow arrowheads indicate the PSD95 signal (green) contiguous to the Syn1 signal (Magenta) on MAP2-positive fibers. Scale bar, 10 µm. Similar results were obtained in three independent experiments.

**Figure 2 ijms-24-12605-f002:**
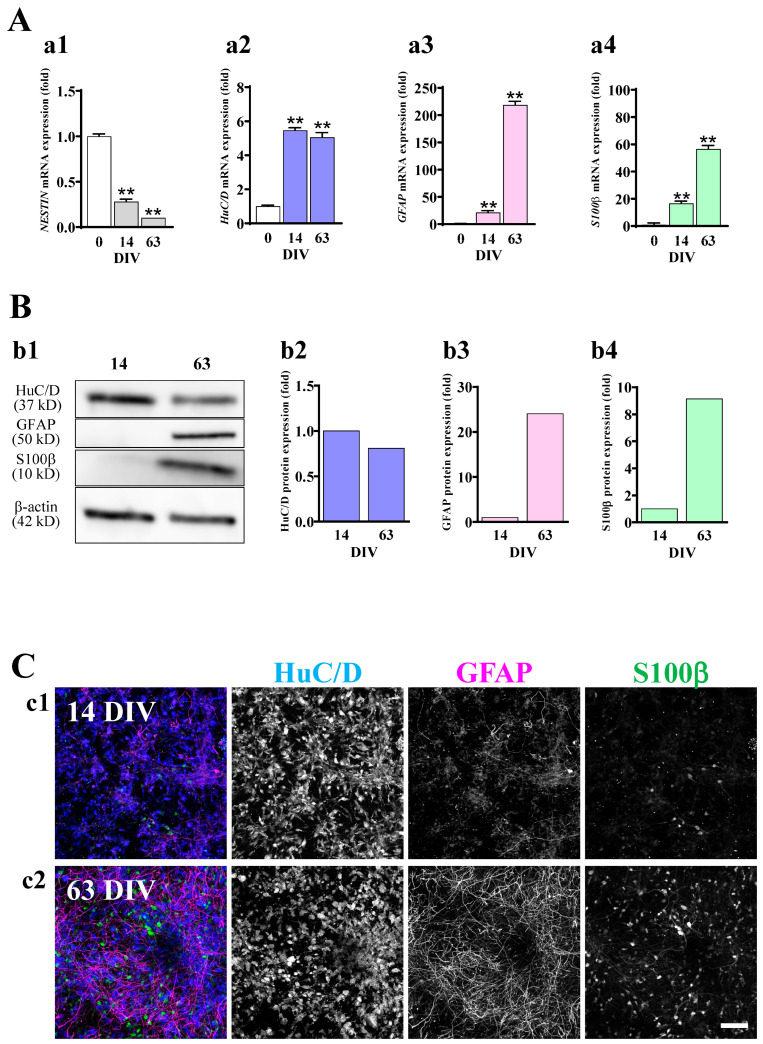
Astrocytes are differentiated in hiPSC-derived neural culture. (**A**) The significant decrease in the mRNA expression level of *NESTIN* (**a1**) and increase in the expression of *HuC/D* (**a2**), *GFAP* (**a3**), and *S100β* (**a4**) along with culture days were confirmed by qRT-PCR (*n* = 3). ** *p* < 0.01 vs. DIV 0 group by Tukey’s test following one-way ANOVA. Data are expressed as the means ± standard deviations. Similar results were obtained in three independent experiments. (**B**) Representative immunoblot at 14 and 63 DIV (**b1**). The expression level of each marker was normalized to 14 DIV. HuC/D protein tended to decrease with culture days (**b2**). GFAP (**b3**) and S100β (**b4**) were increased over time. Similar results were obtained in four independent experiments. (**C**) hiPSC-neurons were immunostained with anti-HuC/D (blue), anti-GFAP (magenta), and anti-S100β (green) antibodies at 14 (**c1**) and 63 DIV (**c2**). Scale bar, 100 µm. Similar results were obtained in three independent experiments.

**Figure 3 ijms-24-12605-f003:**
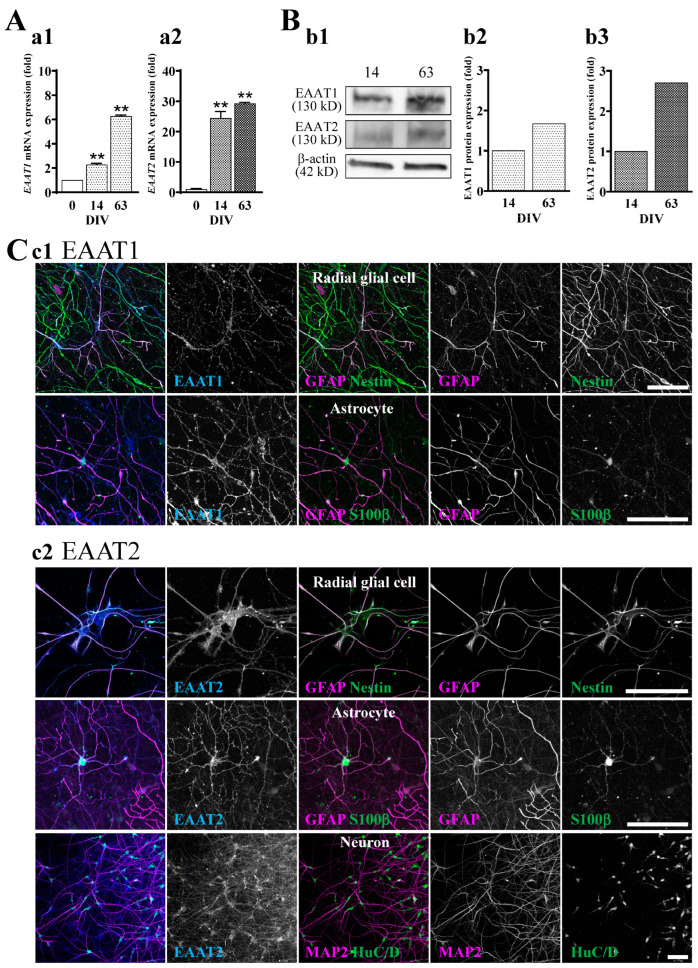
Expression of EAAT1 and EAAT2 in hiPSC-derived neural cells. (**A**) The significant increase in the mRNA expression levels of EAAT1 (**a1**) and EAAT2 (**a2**) along with culture days was confirmed by qRT-PCR (*n* = 3). ** *p* < 0.01 vs. DIV 0 group, Tukey’s test following ANOVA. Data are expressed as the means ± standard deviations. Similar results were obtained in three independent experiments. (**B**) Representative immunoblot at 14 and 63 DIV (**b1**). The expression level of each marker was normalized to 14 DIV. The expression levels of EAAT1 (**b2**) and EAAT2 (**b3**) protein tended to increase with culture days. Similar results were obtained in four independent experiments. (**C**) Identification of cell types expressed EAAT1 and EAAT2 at 63 DIV. We used the following cell markers: GFAP+Nestin+ for radial glial cells, GFAP+S100β+ for astrocytes, and HuC/D+ or MAP2+ for neurons. (**c1**) EAAT1 was localized in radial glial cells (top) and astrocytes (bottom). (**c2**) EAAT2 was localized in radial glial cells (top), astrocytes (middle), and neurons (bottom). Scale bar, 100 µm. Similar results were obtained in three independent experiments.

**Figure 4 ijms-24-12605-f004:**
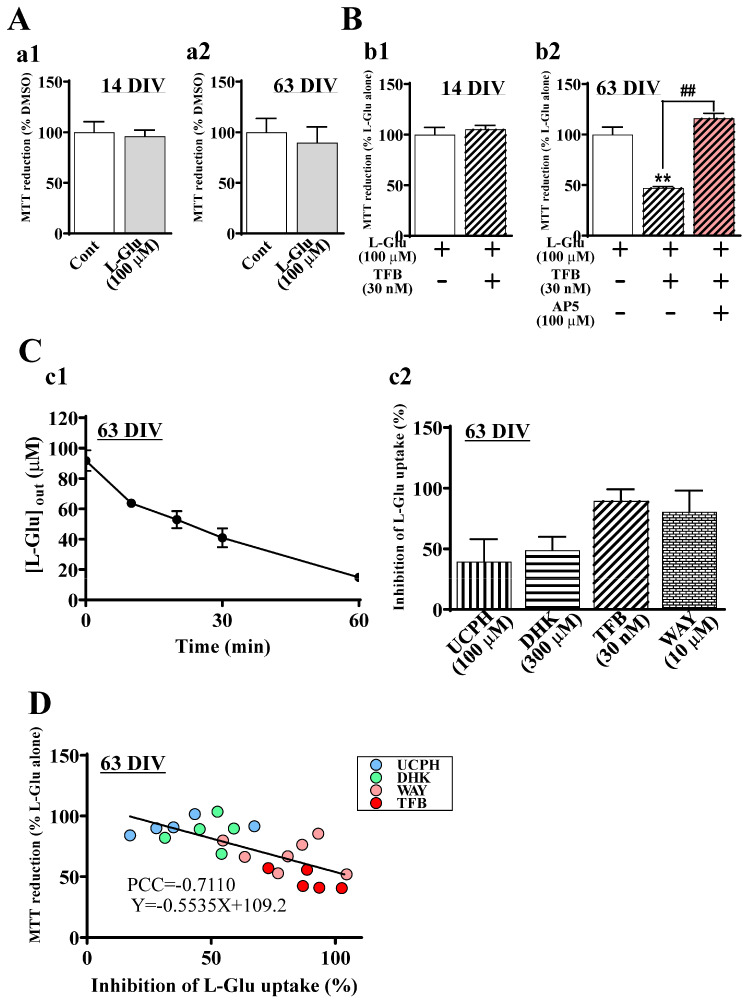
Inhibition of EAATs increases the extracellular L-Glu concentration and leads to excitotoxicity at 63 DIV. (**A**) The effect of L-Glu alone on cell viability at 14 and 63 DIV (*n* = 6). Cell viability was assessed using the MTT reduction assay. At 14 (**a1**) and 63 (**a2**) DIV, the application of L-Glu at 100 μM for 24 h did not change MTT reductions compared with the application of DMSO at 0.4% (Cont), which was used as a vehicle. Unpaired *t*-test. Data are expressed as the means ± standard deviations. Similar results were obtained in three independent experiments. (**B**) The effects of TFB-TBOA (TFB, nonspecific EAAT inhibitor, 30 nM) when coapplied with L-Glu on cell viability at 14 and 63 DIV (*n* = 4–6). At 14 DIV, TFB caused no effect on MTT reductions (**b1**). On the other hand, at 63 DIV, TFB significantly decreased MTT reductions, and AP5 (NMDAR antagonist, 100 μM) blocked the decrease in MTT reductions by TFB (**b2**). Similar results were obtained in three independent experiments. ** *p* < 0.01 vs. L-Glu(+)TFB(−)AP5(−) group by Tukey’s test following ANOVA. ^##^
*p* < 0.01 vs. L-Glu(+)TFB(+)AP5(−) group by Tukey’s test following ANOVA. Data are expressed as the means ± standard deviations. Similar results were obtained in three independent experiments. (**C**) Identification of the contribution of specific EAATs to the decrease in exogenously applied L-Glu. (**c1**) Change in the concentration of L-Glu in the medium ([L-Glu]_out_) after L-Glu (100 μM) was applied at 63 DIV (*n* = 3). [L-Glu]_out_ was nearly zero at 60 min. Data are expressed as the means ± standard deviations. Similar results were obtained in three independent experiments. (**c2**) The effects of EAAT inhibitors on L-Glu uptake at 63 DIV (*n* = 5–7). The effects of EAAT inhibitors on the decrease in [L-Glu]_out_ were assessed at 30 min, which was the time for the 50% decrease in [L-Glu]_out_ from Figure 4C(c1). The percentage inhibition of EAAT inhibitors on L-Glu uptake activity was calculated as 100% of the decrease in extracellular L-Glu concentration in the absence of EAAT inhibitors. UCPH-101 (UCPH, EAAT1 selective EAAT1 inhibitor, 100 μM), dihydrokainic acid (DHK, a competitive EAAT2 inhibitor, 300 μM), WAY213613 (WAY, 10 μM), or TFB inhibited L-Glu uptake. Data are expressed as the means ± standard deviations. (**D**) Inverse correlation between the strength of L-Glu uptake inhibition and the cell viability caused by EAAT inhibitors. The Pearson’s correlation coefficient (PCC) = −0.7110. EAAT inhibitors are shown as follows: UCPH: blue dot; DHK: green dot, WAY: pink dot, and TFB: red dot.

## Data Availability

All data are contained within the manuscript.

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
