# Peer review of "Neuroprotective Potential of L-Glutamate Transporters in Human Induced Pluripotent Stem Cell-Derived Neural Cells against Excitotoxicity"

_ijms, 2023, doi:10.3390/ijms241612605_

Round 1

Reviewer 1 Report

The authors in this study use hIPSC-derived neural cells in an effort to establish a better model for drug development. Furthermore, they analyze the function of L-Glu transporters. They identified EAAT1 and EAAT2 are the dominant transporters that regulate the uptake of L-Glu.

Overall, while this study is of great interest, a more detailed analysis should be provided. Some of the things that must be addressed are:

1. The establishment of the cells must be more detailed. Here they only show the expression of few markers. Since they focus on utilizing these cells in preclinical studies, the establishment of the cell type needs to be compared with already established primary and immortalized cell lines, as well as in vivo analysis. Using the comparison (RNAseq, morphology, markers...), then they should focus on the differences that are indicated in the manuscript.

2. Use different stem cells to achieve the same goal. Either other iPSCs, or embryonic derived neural cells. Showing this study in isolation does not have enough impact.

Additional minor comments that need to be addressed are:

3. Indicate why different timepoints were used for different experiments. First the authors use day 14 and 63, then they shift to 28 and 56. No clear reason is given for any of these timepoints. If possible the timepoints must be consistent and in all experiments use the same days.

4. Please indicate what microscopy was used for imaging. This can either be in Material and Methods, or in the results section.

5. Fig2-B needs to be changed. Higher magnification is necessary for clear co-localization. Either in the form of insert panels or as separate images.

6. Fig3-D is a great experiment, however more detailed analysis is needed. Using the inhibitors, perform a timecourse analysis where you would add L-Glu and add the inhibitor at different timepoints. Is there a difference to when the inhibitor is applied? Maybe initially EAAT1 is important but later EAAT2 comes into play?

7. Fig 4-C. For better clarity, indicate the proposed cell types in the figure. (not only description).

8. 4C-c2. S100b and the merge staining are inconsistent. There are two green dots in the merged panel (bottom right) but those are not as prominent in the individual green channel.

9. In the Discussion section: The significance of our data for preclinical studies. The last sentence is poorly constructed. Instead of avoiding, we should encourage discussing the difference between models even more. That way we can know why there are severe differences and we could address them in future studies.

Reviewer 2 Report

The paper looked at the role of L-glutamate transporter neuroprotective effects against excitoxicity in iPSC derived human neural cells. They used a commercially available iPSC neural cell line. Using PCR, Western blots, immunohistochemistry, they showed EAAT1 and 2 protected these cells in culture from L-glutamate toxicity. 

Recommendations: 

1.     Please consider revising your introduction to clearly define your goals and hypothesis. Currently it is not clear and why you are doing the experiment. 

2.     Additionally, please add more information to the introduction. In your introduction there needs to be a better transition to discussing astrocytes and why the reader should be interested. 

3.     Please make sure all your figure legends are consistent. They should all state the number of animals per group, the statistical test used, and that the error bars are standard deviation. Some legends have parts of the information but not all of it. 

4.     Please add error bars and significant stars to figure 1, panel b2, b3, and b4. If you cannot add error bars and statistical tests because of low animal numbers, please add additional animals. 

5.     Please indicate why there is no Western blot or immunohistochemistry data for MAP2 or PCR data for HuC/D. either include the data or indicate why these data was not included. 

6.     Please add significant stars to figure 2, panel a3, a4, a5, and a6. Please make the yellow arrow heads larger. The arrow heads in b2 should be a different color since they are indicating a different colocalization, please change the color, enlarge them, and update the figure legend. Please include a high magnification of the of the area you are indicating for both b1 and b2. The current images are low magnification, and it is difficult to see the details you are trying to indicate. Additionally, please include higher resolution images for both the low and high mag images. 

7.     What analysis was done to detect colocalization of the proteins in the immunostaining. Please add that information to the method section. If no quantitative analysis was done, please add it as a limitation in the discussion section. 

8.     Please include error bars for figure 4 b2 and b3. If you cannot add error bars and statistical tests because of low animal numbers, please add additional animals. 

Reviewer 3 Report

In this manuscript, Kanako Takahashi  et al studied the L-glutamate transporters distribution and their functions in human pluripotent stem cell derived neural cells. This is an interesting study, as the mechanism of hiPSC derived neural cell exhibiting large tolerance to excitotoxicity was less explored. There is a few concerns before we move on.

1. The authors differentiated iPSC into astrocyte and neuron, and presented nice immunofluorescence images, but i still not sure about the exact identity of these neurons, are they cortical neuron? MAP2 is one general neural marker.

2. According to the author, EAATs have 5 subtypes, EAAT1 and EAAT2 was detected in hiPSC derived astrocyte and neuron; how about the expression of other 3 subtypes in these cells?

3. The author showed that their neural culture include both astrocyte and neuron, and the number of astrocyte increase with the culture time; they also showed that EAAT1 and EAAT2 play main roles in preventing excitotoxicity in this neural culture. As there are cross talk between astrocyte and neuron, what is the exact contribution of astrocyte and neuron to the tolerance, any synergistic effect?      

4. In presenting the data, to present the data of "The expression of EAAT1 and EAAT2 in hiPSC-derived neural cells" before "functional study" be more acceptable for the reader?  

NA

Round 2

Reviewer 2 Report

Thank you for addressing my comments.